# Polysaccharides-Calcium Phosphates Composite Beads as Bone Substitutes for Fractures Repair and Regeneration

**DOI:** 10.3390/polym15061509

**Published:** 2023-03-17

**Authors:** Florina-Daniela Cojocaru, Ioannis Gardikiotis, Gianina Dodi, Aurelian Rotaru, Vera Balan, Elena Rezus, Liliana Verestiuc

**Affiliations:** 1Biomedical Sciences Department, Faculty of Medical Bioengineering, Grigore T. Popa University of Medicine and Pharmacy of Iasi, 9-13 Kogalniceanu Street, 700454 Iasi, Romania; 2Advanced Research and Development Center for Experimental Medicine (CEMEX), Grigore T. Popa University of Medicine and Pharmacy of Iasi, 9-13 Kogalniceanu Street, 700454 Iasi, Romania; 3Department of Electrical Engineering and Computer Science, Stefan cel Mare University of Suceava, 720229 Suceava, Romania; 4Department of Rheumatology and Physiotherapy, Faculty of Medicine, Grigore T. Popa University of Medicine and Pharmacy of Iasi, 700454 Iasi, Romania; 5I Rheumatology Clinic, Clinical Rehabilitation Hospital, 700454 Iasi, Romania

**Keywords:** sodium alginate, guar gum, calcium phosphates, composite beads, bone substitutes, fractures

## Abstract

The tendency of population aging is continuously increasing, which is directly correlated with a significative number of associated pathologies. Several metabolic bone diseases such as osteoporosis or chronic kidney disease–mineral and bone disorders involve a high risk of fractures. Due to the specific fragility, bones will not self-heal and supportive treatments are necessary. Implantable bone substitutes, a component of bone tissue engineering (BTE) strategy, proved to be an efficient solution for this issue. The aim of this study was to develop composites beads (CBs) with application in the complex field of BTE, by assembling the features of both biomaterials’ classes: biopolymers (more specific, polysaccharides: alginate and two different concentrations of guar gum/carboxymethyl guar gum) and ceramics (more specific, calcium phosphates), in a combination described for the first time in the literature. The CBs prepared by double crosslinking (ionic and physically) showed adequate physico-chemical characteristics and capabilities (morphology, chemical structure and composition, mechanical strength, and in vitro behaviour in four different acellular simulated body fluids) for bone tissue repair. Moreover, preliminary in vitro studies on cell cultures highlighted that the CBs were free of cytotoxicity and did not affect the morphology and density of cells. The results indicated that the beads based on a higher concentration of guar gum have superior properties than those with carboxymetilated guar, especially in terms of mechanical properties and behaviour in simulated body fluids.

## 1. Introduction

Metabolic bone diseases (MBDs) include a significant spectrum of clinically related disorders, caused by calcium, phosphorus, magnesium or vitamin D abnormalities. The most common triggers for MBDs are hereditary hypophosphatemia, hyperparathyroidism or a deficiency of vitamin D. Osteoporosis, osteomalacia (with the childhood equivalent, rickets) and fluorosis are the most frequently diagnosed, while Paget’s disease, chronic kidney disease-mineral bone disorder, fibrous dysplasia and osteogenesis imperfecta are rarely met [1,2].

Usually, the patients with MBDs do not experience symptoms, evidence of illness or abnormality, until they develop disastrous musculoskeletal complications, such as fractures [3]. Fractures have an enormous human and socio-economic impact, for example, in case of osteoporosis almost nine million fractures every year are reported, given the fact that every 3 s an osteoporotic bone will break [4]. Since osteoporosis mainly affects older persons [4] and due to the aging of the global population, it is anticipated that its incidence and thus the number of fractures will double in the next 20 years [5]. Human bone is a dynamic tissue, 5 to 10% of the skeleton being replaced annually in the case of an adult [6], in healthy conditions due to a perfect balance between bone resorption, mediated by osteoclasts and bone formation, facilitated by osteoblasts [7]. As the skeleton ages and due to several pathologies, such as MBDs, significant physiological and structural changes occur [8], and the bone needs support for proper repair.

Bone tissue engineering (BTE) is an intensely studied area that offers promising solutions for treating bone disorders, especially for the reconstruction of serious bone defects, such as those caused by fractures [9]. Implantable bone substitutes (BS), a significant component of BTE strategy, offer structural support for cells and newly formed bone tissue and sustain the natural processes of tissue regeneration and development [10]. There are two main classes of BS, derived from biological and synthetic products. Demineralized bone matrix, platelet-rich plasma, bone morphogenetic proteins, hydroxyapatite and coral enter the natural category, while calcium sulphate, calcium phosphate cements, biphasic calcium phosphates, calcium phosphate ceramics, bioactive glasses and polymer-based substitutes, constitute the synthetic BS. Polymer-based BS refers frequently to ceramic/polymer composites, that provide combined features, the toughness of the polymer and rigidity of the ceramic [11].

Naturally occurring polymers are among the most used biomaterials in bone substitutes manufacturing. Polysaccharides dominate the top-level position in this application, mainly due to their easy availability, eco-friendliness, and non-toxicity. Guar gum (GG) and alginate (ALG) are two polymers widespread in nature, obtained from cheap naturally available and affordable sources [12].

Due to its glycopolysaccharide structure, GG, isolated from ground endosperm of Cyamopsis tetragonolobus or Cyamopsis psoraloides, interacts and binds with proteins in a physiological environment [13]. Carboxymethyl GG (CMGG) expresses increased hydrophilicity and clarity solution [14] and can be designed as nanoparticles and microspheres to hydrogels for various biomedical applications [15].

ALG, another vegetal polysaccharide, isolated from brown algae, is frequently chosen in tissue engineering in general, and in bone tissue engineering, based on its remarkable properties: biocompatibility, biodegradability, injectability and capacity to support cell growth. Moreover, in the presence of divalent cations (Ca^2+^) it generates a stable gel [16].

Besides polymers, calcium phosphates, represented by hydroxyapatite are found in the composition of ceramic/polymer bone substitutes, because of their osteogenicity and osteoconductivity. But, due to hydroxyapatite’s brittle nature it is difficult to shape and handle it; therefore, it is often combined with different polymers, either natural or synthetic [17].

ALG-ceramic biomaterials have already proved their great merits for use in the bone tissue engineering field, the results of hundreds of studies being synthetized in a significant number of reviews [18,19,20,21], but the combination between GG and ceramics is limited to only a few studies. Nayak et al. [22,23], prepared hydroxyapatite-ciprofloxacin composites using GG as a binder, for application as bone implants, characterized in two studies. In another study, guar gum was used together with konjac glucomannan to improve the inferior anti-washout property of injectable calcium phosphate cement [24]. As a gel-forming galactomannan, GG was mixed with gellan gum and hydroxyapatite, crosslinked with sodium trimetaphosphate, finally becoming obtainable in 3D freeze-dried scaffolds suitable for bone tissue engineering applications [25]. In a significant study published by Senthilarasan et al., the authors highlighted the antibacterial and anti-inflammatory activity of nano-hydroxyapatite/GG composites [26]. No study has been found regarding the combination of CMGG and ceramics, but what is important is the fact that GG and CMGG can also be crosslinked in situ by Ca^2+^ ions [27], so can be mixed with ALG to prepare composites crosslinked with CaCl_2_. Based on our knowledge, the mixture of ALG with GG or CMGG and calcium phosphates has not yet been reported in the literature.

Considering all these aspects, the aim of the present study was to develop composite beads based on ALG, GG or CMGG, and calcium phosphates by the ionic crosslinking method, to be further used as implantable bone substitutes.

The basic properties that a bone substitute should possess, in order to be successfully implanted in bone defects caused by grave fractures are: chemical stability in body fluid, adequate biomechanical support, cost-effectivness [28], specific internal structure (crystallinity, pore size and distribution), surface topography, biodegradability/resorbability, biocompatibility [29], and easy handling [30]. The beads obtained will be characterized in terms of physico-chemical properties. First, their external and internal morphology will be studied, as well as their composition and chemical structure, mechanical features, and interaction with simulated body fluids. The last assays will be related to preliminary in vitro cell culture behaviour.

In a future study, the key features of a bone graft will also be analysed; these are osteoconduction, osteoinduction, osteogenesis, and structural support [31].

## 2. Materials and Methods

### 2.1. Materials

For the preparation of the beads, the following were used: calcium phosphate tribasic, 34–40% Ca (HA—empirical formula: Ca_10_(OH)_2_(PO_4_)_6_—approximate, Alfa Aesar, Kandel, Germany); alginic acid, sodium salt (ALG—Sigma-Aldrich, Drammensveien, Norway); guar gum (GG—Sigma-Aldrich, Karachi, Pakistan) or carboxymethyl guar gum (CMGG—prepared according to the protocol described first by Dodi et al. [32]); CaCl_2_ dihydrate (Sigma-Aldrich, Steinheim am Albuch, Germany).

The reagents used to study in vitro interaction with simulated body fluids were: a simulated body fluid (SBF) prepared following a protocol described in a previous article [33]); physiological sodium chloride solution, NaCl 0.9% (B. Braun, Melsungen, Germany); Dulbecco’s Modified Eagle’s medium (DMEM), high glucose, with glutamine and Hanks’ Balanced Salt Solution (HBSS), no calcium, no magnesium, no phenol red (DMEM and HBSS were from Biological Industries, Beit-Haemek, Israel).

In vitro cell behaviour assays were performed using the same DMEM and HBSS mentioned above; HBSS with calcium and magnesium, no phenol red; fetal bovine serum (FBS); antibiotics mixture (penicillin-streptomycin-neomycin mixture—P/S/N, ~5000 units penicillin, 5 mg streptomycin and 10 mg neomycin/mL); Thiazolyl Blue Tetrazolium Bromide (MTT); Dimethyl sulfoxide (DMSO) and Calcein-AM solution, 4 mM in DMSO, min. 90% (HPLC). Apart from HBSS, which was purchased from Biological Industries, Beit-Haemek, Israel, all the other reagents were purchased from Sigma-Aldrich, Steinheim am Albuch, Germany.

All the solutions used in this study were prepared with type 1 ultrapure water from Arium^®^ Mini Ultrapure Water Systems (Goettingen, Germany).

### 2.2. Composite Beads Preparation

The composite beads (CBs) were obtained by ionic crosslinking of ALG, GG and CMGG with CaCl_2_, in the presence of HA. Inspired by the composition of the human bone [34], the CBs include 33% wt. organic compounds (biopolymers) and 67% wt. inorganic compounds (HA). The first step was the mixture of the polymers: ALG, GG or CMGG and HA. For a proper homogenization, the mixture was ultrasonicated, 30 s (UP50H homogenizer, Hielscher Ultrasonics, Teltow, Germany, amplitude 100%), then dropwised into a CaCl_2_ bath (0.4 M) and left to crosslink for 2 h at room temperature. After repeated washing procedures with ultrapure water, the beads were dried using a halogen moisture analyzer (HC103/01, Mettler Toledo, Greifensee, Switzerland), at a drying temperature of 105 °C.

### 2.3. Composite Beads Characterization

#### 2.3.1. Morphology and Composition

The microstructural features of the beads and their compositions were examined using the Hitachi SU-70 Field Emission Scanning Electron Microscope (FE-SEM) with Oxford Energy-dispersive X-ray Spectrometer (EDX) (Hitachi, Buckinghamshire, UK). Before examination, each bead was cross-sectioned and covered with a thin platinum layer. The working parameters for FE-SEM-EDX were: 20 kV accelerating voltage, working distance between 16.8 and 17.6 mm, 30° elevation and 20 keV energy level.

#### 2.3.2. Chemical Structure

CBs’ chemical structure was analysed using Fourier-transform infrared spectroscopy with attenuated total reflection module (ATR–FTIR), on Nicolet Summit Pro FTIR Spectrometer with Everest ATR accessory, from Thermo ScientificTM, Waltham, MA, USA. The spectra were collected at the following parameters: average of 16 scans between 4000 to 400 cm^−1^, at a 4 cm^−1^ resolution.

#### 2.3.3. Mechanical Features

CBs’ mechanical features were studied by an axial compression test, performed on a texture analyzer (TA) device—TA-XT2 Plus (Stable Microsystems, Godalming, UK), equipped with a 5 kg load cell.

Before each analysis, the bead diameter was measured with a digital caliper (mean diameter was 1.7 ± 0.1 mm), and then placed in the center of the TA-XT2 sample platform. To better understand the interaction between organic and inorganic compounds, from a mechanical point of view, pure HA was also analysed. For this, HA discs (0.16 ± 0.01 g, diameter 13 mm) were prepared using CrushIR™ digital hydraulic press (Pike Technologies, Madison, WI, USA) by applying a force of 7.6 tons.

TA-XT2 is connected to a computer, the commands and the results being given/displayed using a specific software (Exponent). For each analysis, the following parameters were used: 1 mm/second pre-test speed, 2 mm/second compression speed, and 0.2% deformation (ε). The device displays a force-time curve at a specific ε, expressed in g (1 g = 0.0098 N). This force, beads/HA discs diameter and ε were used to determine the compressive strength—σ (MPa) and Young’s Modulus—E (MPa), using Hooke’s law. Four aliquots from each CBs and four HA disks were tested, the results being expressed as mean ± standard deviation, Figure A6. highlights the schematic representation of the Texture Analyser (TA-XT2 Plus) setup for beads mechanical studies.

#### 2.3.4. In Vitro Interaction with Simulated Body Fluids

Four different acellular aqueous media were chosen to study CBs’ in vitro interaction: DMEM (the same used for cell culture assays), HBSS, SBF and physiological sodium chloride solution 0.9%. Beads with an initial weight of 3–4 mg were immersed in 1 mL of each of the four mentioned solutions, in a 2 mL Eppendorf microcentrifuge tube and placed at 37 °C, in Eppendorf^®^ ThermoMixer C with ThermoTop lid, used to prevent condensation on tube lid and tube wall (Hamburg, Germany), without shaking. In all the cases, the physiological medium was refreshed after each measurement (daily for the first 4 days and then weekly).

The gravimetric method was used to determine the swelling degree (*SD*) of the beads based on *wo* (the initial weight of beads) and *we* (the weight of the bead after immersion, at specific time intervals):(1)$$SD=\frac{we-wo\;}{wo}\times 100\;\left(\%\right)$$

The study was developed in triplicate for each CB and the results were displayed as the mean ± the standard deviation.

#### 2.3.5. In Vitro Cell Behaviour

##### MTT Assay

The MTT assay was performed separately on two different cell lines: MG63 human osteosarcoma cells (ATCC, Rockville, MD, USA) and A431 human epidermoid carcinoma cells (Cell Service, Eppelheim, Germany), after the approval of the Ethical Committee of Grigore T. Popa University of Medicine and Pharmacy of Iasi, Romania (no. 141 from 26 January 2022). A standard protocol, already described by our group in another published paper, was used [33]. In brief, Mg-63 and A431 were incubated for 24 h in conditions proper for cell cultures (5% CO_2_, 37 °C, 95% relative humidity), in complete DMEM enriched with 10% FBS and 1% P/S/N, in plates of 48 wells (10 × 10^3^ cells/well). After a proper sterilization (UV exposure for 2 h, 1 h on each part), the beads were incubated with the already attached cells (direct contact). Considering the size of the beads, and their interaction with DMEM and HBSS, 48 well plates were used and the MTT test was accomplished for four aliquots (1 bead/well, mean weight 3.5 ± 0.3 mg and mean diameter around 1.7 ± 0.1 mm) at 24, 48 and 72 h.

##### Calcein-AM Assay

The Calcein-AM assay was carried out after 72 h of direct contact of the beads with MG63 cells, as CBs’ main application is bone tissue engineering. In brief, after removing the beads and the medium from the wells, the cells were washed two times with HBSS with calcium and magnesium. A volume of Calcein-AM solution prepared in HBSS (2:1000) was added in each well to completely cover the cell layer and was maintained for 40 min in incubator, at specific conditions for cell cultures. Inverted Phase Contrast Microscope (Leica, Wetzlar, Germany), 10× objective, was used to study cells morphology and density.

## 3. Results and Discussions

### 3.1. Composite Beads Preparation

CBs based on ALG, GG/and HA were prepared by ionic crosslinking, known as a simple, common and cost-effective gelation method to form alginate-based materials [35]. The preparation flowchart is depicted in Figure 1. After crosslinking and purification steps, the obtained composite mixtures were processed into their final form at 105 °C, using a halogen moisture analyzer. This increased drying temperature was inspired by the Dehydrothermal (DHT) treatment, a physical crosslinking method based on temperatures above 90 °C and vacuum used to remove water from polymer molecules and to induce intramolecular crosslinks [36,37]. The halogen radiator from the moisture analyzer, very quickly reached the chosen operating temperature and kept it accurately during the process. Moreover, compared with the classic oven drying, it offered superior heat distribution throughout the sample and a shorter drying period [38]. GG and CMGG were used separately in order to evaluate whether the water-soluble form of GG, which is easy to handle and homogenously distributed in the polymeric composition, determine the enhanced properties of the beads, compared with those based on bare gum that is partially soluble in water.

These two concepts, CaCl_2_ crosslinking and high drying temperature, were chosen in order to obtain beads with adequate mechanical strength and suitable handling properties, which can sustain fractures repair and regeneration. To the best of our knowledge, using a halogen moisture analyser to design composite beads has not yet been reported in the literature. Four different CBs in terms of biopolymers content were obtained, their codification and composition being detailed in Table 1.

### 3.2. Morphology and Composition

Figure 2 presents the morphology of the synthesized beads and the microstructure of their cross-sections, at a magnification between ×40 and ×50 (for a general view), and at high magnifications (between ×350 and ×25,000) to observe reasonable details that can influence CBs’ features and biological properties.

On a 1 mm scale, the FESEM micrographs exhibited beads with micro spherical shape, a compact structure, and rough surface, with no evidence of cracks. The roughness of the CB surface is useful because it improves cells’ attachment, proliferation and differentiation [39].

As the degree of magnification increased (between ×350 and ×1100), it was observed that the HA microparticles are uniformly distributed in the polysaccharide’s matrix, attributed to the ultrasonic homogenization used in the preparation process. Also, the inorganic components of the beads were well embedded and densely packed in the organic phase [40]. The HA microcrystals can be compared to some spherulites or filaments, suggesting a rugged coral reef-like surface, denoted by Liu et al., as a specific pattern for hydroxyapatite [41].

At a magnification between ×13,000 and ×25,000, some cracks were observed in the beads structure that frequently appear in the drying process of the crosslinked alginate-based materials [42] and is attributed to the polymeric matrix contraction under the temperature effect.

Regarding the porosity of the CBs, Figure 2, Figure 3 and Figure 4, Appendix A, Appendix A, Appendix A and Appendix A (Appendix A), show, in all the cases, interconnected microparticles interspersed with microporous and macroporous structures. By comparing the micrographs of the beads containing GG (CB1 and CB2) and those with CMGG (CB3 and CB4), fluctuating porosities and different distributions of ceramics in the organic part can be detected. The surface architecture and porosity of the GG beads is similar to that of the Bio-Oss™ xenogeneic bone substitute, characterized in a study by de Oliviera et al. [43], where the authors remarked that a superior homogenicity and a better coalescence existed between the particles, in comparison with the biphasic material Nanosynt^®^ (60% hydroxyapatite and 40% β-tricalcium phosphate), whose structure can be associated with that of CMGG beads. As an important conclusion of the study, the authors highlighted the fact that both porous structures sustained bone ingrowth on their surfaces and inside them after their implantation in rat calvaria [43]. Moreover, the porous structure of CMGG beads can also be compared with that of micro–macro biphasic calcium phosphate bone graft, presented by Shiu et al. [44].

These differences between GG beads and CMGG beads, in terms of porosities and ceramic distribution in the organic part, are explained by the results of a study published by Dalei et al. [15], where GG micrographs revealed a discrete and elongated structure with particles separated from each other, while CMGG micrographs showed a substantially altered morphology, with particles adhering to each other and forming larger aggregates, more evidence of an increased CMGG concentration.

EDX is a valuable microanalysis used in various biomedical fields; with regard to the specimens containing calcium phosphates, it is considered a semi-quantitative method, since it allows the determination of Ca/P molar ratio, indicating the nature of the calcium phosphates located close to the matrix surface [45]. The elemental mapping results of the four CBs (Figure 3a,c, Figure 4a,c and Appendix A, Appendix A, Appendix A and Appendix A) proved the existence of the expected elements, namely C, O, Ca, P, and Cl, suggesting that HA were uniformly distributed in the polymeric matrix. Table 2 and Figure 3b,d and Figure 4b,d, present the result of the elemental analysis, used to determine the Ca/P ratio.

In the case of calcium phosphate bioceramics for bone reconstruction, due to natural bone composition biomimetics, the Ca/P ratio significantly influenced CBs in vitro interaction with osteoblast. For instance, it was reported that a Ca/P ratio value higher than 2 decreased osteoblast viability; nevertheless, high osteoblast alkaline phosphatase activity was identified for the bioceramics with the Ca/P ratio between 0.5–2.5 [46]. Moreover, the Ca/P ratio strongly influences the mechanical properties of the biomaterials [47], thereby contributing to better understanding of the features of the obtained beads.

According to the producer data, the HA incorporated in the beads structure have the estimated chemical formula Ca_10_(OH)_2_(PO_4_)_6_, the molecular weight of 1004.67 g/mol, which is known to have a theoretical Ca/P ratio of 1.67 [48]. Generally, the compounds with a Ca/P molar ratio between 1.2 and 2.2 are defined as amorphous calcium phosphate (ACP), such as those found for CB2 and CB3. In particular, if the value of Ca/P ratio is placed in the range 1.5–1.67, it is named calcium deficient hydroxyapatite (CDHA) [49,50]. Ca/P ratio of CB1 fits in the range of 0.5–2.5, mentioned above, so a proper in vitro behaviour is anticipated for this bead. The ratio for CB4, where the calcium quantity is 2.7 higher than the phosphorous one, can be attributed to a stronger interaction between the Ca^2+^ ions and the biopolymers functional groups and a high crosslinking degree. Moreover, CB4 showed an irregular porous surface structure with particles aggregated between them. Since EDX offers the results from the surface of the analysed sample rather than from inside of it, the high value of Ca/P can also be related to these aspects.

### 3.3. Chemical Structure

By FTIR spectroscopy, the functional groups of the four CBs and the interactions between components were identified. The wavenumber values for each functional group identified in the polymers, HA and the four CBs, can be seen in Table A1.

On the ATR–FTIR spectra of the four CBs (Figure 5) the absorption bands specific for the used polysaccharides and the calcium phosphates were identified. The peaks around 1600 cm^−1^ and 1420 cm^−1^ are specific to the biopolymers: in the case of ALG representing COO^−^ asymmetric and symmetric stretching [51], while for GG/CMGG indicated the presence of carbonyl groups [52]. Regarding HA, the peaks near 1020 cm^−1^ can be assigned to the vibrational modes of ν_3_ PO_4_^3–^, whereas those at about 600 cm^−1^ and 560 cm^−1^ can be attributed to vibrational modes of ν_4_ PO_3_^4–^ [53].

The high intense peaks identified near 1020 cm^−1^ from all the spectra are specific to the biopolymers (C–C stretching) that overlap those of the HA (ν_3_ PO_4_^3−^) [51]. Also, these peaks can be attributed to a strong O–H vibration and the binding of the Ca^2+^ ions to the guluronic acids [54], suggesting a proper distribution of the HA in the polymeric matrix, also observed from FESEM-EDX data.

The COO^−^ symmetric stretch peak (1593 cm^−1^ in case of ALG, 1637/1639 cm^−1^ for guar gum-based matrix, Figure A5) exhibits a small shift towards 1598/1600 cm^−1^ in the beads’ composition, indicating an ionic binding between the Ca^2+^ and COO^−^ of the polymers [55]. The strong shoulder at 1019/1020/1021/1022 cm^−1^ relating to the C–C and C–O stretching, can also be attributed to the presence of crosslinking [56].

### 3.4. Mechanical Features

Since human bone is constantly subjected to mechanical forces, vital for some specific processes such as homeostasis, skeletal formation, resorption and adaption [57], the substitutes implanted in bone defects should possess similar or at least comparable features [58]. TA-XT2 Plus is a versatile device used to study various features of polymeric materials, such as mechanical properties [59], texture profile analysis [60], or mucoadhesive properties [61]. For example, TA-XT2 was used to analyse alginate beads’ behaviour at compressive stress, presented as hardness of the beads [62], breaking force, and Young’s Modulus [63]. The results of the mechanical assay performed on CBs (Figure A7), expressed as compressive strength and Young’s Modulus, are displayed in Figure 6.

The σ values for beads were obtained in the range of 1.23 MPa (CB1) and 1.66 MPa (CB2), comparable with those of human trabecular bone, placed between 0.1 and 16 MPa, according to the comprehensive review of Gerhardt and Boccaccini [64]. Moreover, the obtained values are similar to those of porous hydroxyapatite bioceramics, defined between 2 and 11 MPa by Prakasam et al. [65], observing that a Ca/P ratio higher than 1.67 results in a decrease of compressive strength. Moreover, it is already known that the stoichiometric hydroxyapatite has the higher value of Young’s Modulus, related to nonstoichiometric HA [66].

This observation is relevant for our study, because as EDX data showed, the only bead with a Ca/P molar ratio smaller than 1.67 (1.625) is CB2 which is the sample with the highest σ. Regarding the compressive or elasticity modulus, the smallest value obtained was 6.18 MPa (CB1), while the highest was 8.32 MPa (CB2). E of CB2 is comparable to that found by Ji et al. (9.5–12.4 MPa) for other bone tissue engineering scaffolds [67].

The suitable values of σ and E obtained at the mechanical assays, comparable with those of human trabecular bone and other ceramics, are assigned to the intermolecular crosslinking [35]. As already mentioned in the introduction, as in the case of ALG, GG and CMGG are also crosslinked in situ by Ca^2+^ ions [27], thus resulting in double network polymeric beads with extra hydrogen bonds formed in the crosslinking process. Li et al. [68] prepared and characterized double-network beads, based on ALG and ĸ-carrageenan, instated of GG/CMGG, and they observed that this kind of network helped improve crosslinking density and mechanical strength.

Even if is generally believed that a high degree of crosslinking involves superior mechanical properties, Montoya-Ospina et al. [69] recently proved that a high crosslinking density is correlated with a decrease in the uniaxial tensile properties, yield strength, stress at break, and displacement at break. Crosslinking, in fact, leads to a decrease in crystallinity and chain mobility, explaining the weak mechanical features.

Regarding pure HA discs, the following values were determined: σ = 0.0913 ± 0.00117 (MPa) and E = 0.4566 ± 0.0058 (MPa) (Figure A8), which are clearly smaller than those obtained for the composite beads. This considerable difference can be explained by ceramic biomaterials lack of ability to plastically deform. More precisely, calcium phosphate ceramics proved to have a specific brittleness and low impact resistance, mainly due to their porosity, microscopic flaws, and micro-cracks, and to their primary ionic bonds. For these reasons, calcium phosphate ceramics are used in dentistry and orthopedy, as non-load bearing implants [48,70,71]. As a solution to these drawbacks, calcium phosphates can be mixed with a polymeric phase, since they offer to the composite biomaterial a higher flaw tolerance [72].

Based on their similarities in terms of composition with natural bone, ceramic/polymer composites are considered superior materials, being part of the third-generation orthopedic biomaterials. They combine the key features of the two classes, resulting in superior materials with great bioactivity, biodegradability, mechanically strong yet at the same time flexible [73].

These properties are governed by the interfacial interactions between organic and inorganic components [74,75]. These interactions can be anticipated using theoretical modelling [75], near-infrared spectroscopy combined with chemometrics [76], high pressure X-ray diffraction, and molecular dynamics simulation [77], among many others, and can be the subject of a separate study due to the complexity of the offered results.

Comparing the beads with GG and those with CMGG, in terms of mechanical features, the results indicated that a high concentration of GG generated the most resistant bead, attributed to the high molecular weight of the polymer.

### 3.5. In Vitro Interaction with Simulated Body Fluids

The first step in the in vitro and in vivo characterization of any material intended to be used in direct contact with the human body, is their immersion in different acellular aqueous environments. Simulated body fluid (SBF), developed by Kokubo et al. [78], was initially introduced to predict in vivo bioactivity of the materials. Furthermore, it is an excellent immersion medium used to anticipate the in vitro and in vivo performance of a biomaterial, since it mimics the inorganic composition of blood plasma. Other appropriate aqueous fluids are Tris-HCl buffer, physiological sodium chloride solution (serum), various cell culture medium formulations, and HBSS [79,80].

In order to get information about how the beads will behave in cell cultures assays, their swelling degree was studied for 14 days, denoted as the mean time for these assays. The obtained results are displayed in Figure 7.

The values attained for SD in DMEM were relatively constant, especially for CB1 and CB2. After seven days of immersion, a slight decrease was noticed for CB2. CB3, and CB4 graphs showed an inconstant behaviour.

Regarding the values obtained for SD in HBSS, CB2, CB3, and CB4 swelled quickly in the first 48 h, followed by an equilibrium phase until day seven, then by a sudden decrease in the value in the 2nd week. The obtained results are specific for the alginate-based materials that are crosslinked with CaCl_2,_ as during the preparation process an egg-box model of crosslinking is created, which is in fact a chelate structure that generated very stable junctions [81]. Moreover, some temporary junctions are created after the interaction of anionic ALG chains with Ca^2+^ ions. The swelling process starts with the diffusion of the water in the bead structure that induces the dissociation of the temporary junctions and the release of Ca^2+^ ions. The polymeric network expands, resulting in an increase of SD, until a critical value is reached; then, the stable chelate junctions are dissociated, and the degradation of the material starts [82].

The second part of the extensive study of beads performance in simulated body fluids was their immersion in physiological serum and SBF for 28 days, with the aim to obtain data concerning their behaviour after implantations in a bone defect, as in vivo experiments request much longer times.

In Figure 8 a constant and gradual increase of SD can be observed in case of CB2 in the first two weeks and a decrease after this period, more evident in case of immersion in SBF. For the beads based on CMGG, a non-uniform swelling was detected for CB4 in both fluids, while for CB3 only in SBF. These irregular values are directly correlated to SEM data, where, as specified before, the micrographs for beads CB1 and CB2 (GG based) had a superior morphology with particles separated from each other and homogenously distributed within the micro-macro porous structure, comparable to CB3 and CB4 (CMGG based), where the particles joined together resulting in large aggregates. This large agglomeration of particles resulted in uneven, porous structures, the consequence being an irregular distribution of aqueous solutions within the structure.

Another important aspect related to CB4, the bead with the highest Ca/P ratio, was the fact that the smallest values for SD among all four aqueous media (Figure 7 and Figure 8) were obtained for it, and as remarked by Annabi et al. [83], porous biomaterials with high degree of crosslinking displayed low swelling ratios.

On the other hand, the highest values of SD were obtained for CB1, the bead with the most modest compressive strength and Young’s Modulus, so is the least resistant from a mechanical point of view. A high degree of swelling is mainly associated with a porous structure, which is directly correlated with low mechanical strength, the balance between an adequate porous structure and superior mechanical properties being a serious challenge for bone tissue engineering materials [84].

In summary, GG formed more stable interactions and junctions in the preparation process compared with CMGG, but future studies are necessary to better understand the encountered mechanism. The bioactivity of the beads after immersion in SBF will be studied in the near feature, as it represents one of the main subjects of another study, where in vitro and in vivo behaviours of CBs will be analysed in detail.

### 3.6. In Vitro Cell Behaviour: Preliminary Data

#### 3.6.1. MTT Assay

Developed first by Mosmann et al. [85] 40 years ago, MTT, which is an abbreviation of the main reagent used in this assay (3-(4,5-dimethylthiazol-2-yl)-2,5-diphe-nyl-2H-tetrazolium bromide), is the most frequent test for the study of cellular metabolic activity. It is based on the ability of the reagent to penetrate first the cell membrane and then the mitochondrial inner membrane, but only in the case of metabolically active cells, where MTT is reduced to a water insoluble product named formazan. Formazan is then solubilized using dimethyl sulfoxide or isopropanol, resulting in a purple solution further studied spectrophotometrically, in terms of its optical density at a specific wavelength (570 nm) [86].

As mentioned in Section 2.3.5, MG63 human osteosarcoma cells and A431 human epidermoid carcinoma cells were used to study the metabolic activity after interaction with CBs, at three different times. In case of direct contact of CBs with osteoblasts, as observed from Figure 9 (left), the values of cells’ viability are relatively constant; therefore, the beads have a non-cytotoxic behaviour in contact with this cell type, since after 72 h the lowest value was 93% and the highest 96%.

To further confirm this feature, the beads were tested in the same conditions in direct contact with A431. Even if the CBs will be used as bone substitutes, their interaction with this tumoral line was studied, mainly as A431 are acknowledged to be highly angiogenic, exhibiting an increased expression of epidermal growth factor receptors and an ability to significantly produce vascular endothelial growth factor, well known for promoting neovascularization and neo-angiogenesis [87].

The interaction of CBs with A431 (Figure 9, right) resulted in the same non-cytotoxic behaviour. CB2 and CB3, the beads with the lowest Ca/P ratio (according to EDX results), showed the most remarkable results, (the cells viability increased in time with 6–7%, from 93.8% at 24 h to 100% at 72 h in case of CB3, and from 96.2% at 24 h to 103.2% at 72 h, in case of CB2). In the case of CB1, the cells’ viability was 88% at 72 h, a suitable result since the non-cytotoxicity is attributed to percentages of cells’ viability higher than 80% [88].

#### 3.6.2. Calcein-AM Assay

Calcein-AM is an easy-to-handle assay that provides useful data on the morphology and density of cells after incubation with compounds/materials intended for biomedical applications. It is a single-staining method based on the enzymatic action of Calcein-AM dye [89].

The results of the assay are presented in Figure 10, which clearly shows that the cells incubated with CBs have the same morphology and density as those from control.

Considering the results obtained from the two preliminary biological in vitro assays, more complex cell culture studies can be safely performed, such as the capacity of the beads to sustain mineralization (e.g., Alizarin Red test) and finally be considered for implantation into animal models.

## 4. Conclusions and Perspectives

Four types of organic-inorganic composite beads were fabricated for the first time by incorporating HA into the ALG-GG/CMGG-based polymer matrix with different contents. The HA microparticles were uniformly embedded and densely packed into the polysaccharide phase by ionic gelation and a dehydrothermal process, forming beads with spherical shapes, compact structures, and rough surfaces. The Ca/P molar ratio determined by EDX was fitted into the amorphous calcium phosphate type (1.2–2.2), calcium deficient hydroxyapatite (1.5–1.67), strong interaction between the Ca^2+^ ions and the biopolymers functional groups (2.35 and 2.7). The use of high drying temperature in the bead preparation, different molar weight polymers, and intermolecular crosslinking determined adequate mechanical strength and suitable handling features that can sustain fractures repair and regeneration. Concentrations of ALG or GG/CMGG and different polymer ratios did not significantly affect in vitro interaction with simulated body fluids and cytocompatibility on the tested cell lines. Thus, the developed HA-ALG-GG/CMGG based beads could be used as a potential bone substitute for bone tissue repair and regeneration. In light of the encouraging results, several perspectives could be adopted, in order to explore an automatic method to obtain optimized beads in terms of weight and dimensions, drug loading, and release studies, as well as complex cell culture assays and in vivo examinations on bone defects.

## Figures and Tables

**Figure 1 polymers-15-01509-f001:**
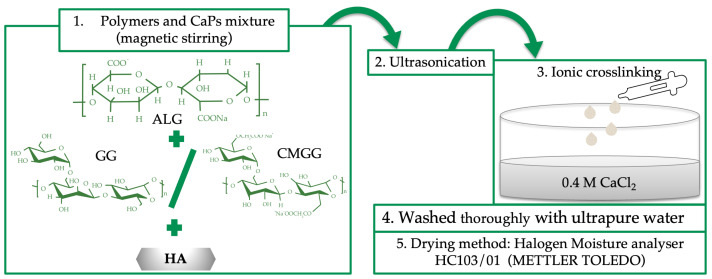
Composite beads preparation flowchart.

**Figure 2 polymers-15-01509-f002:**
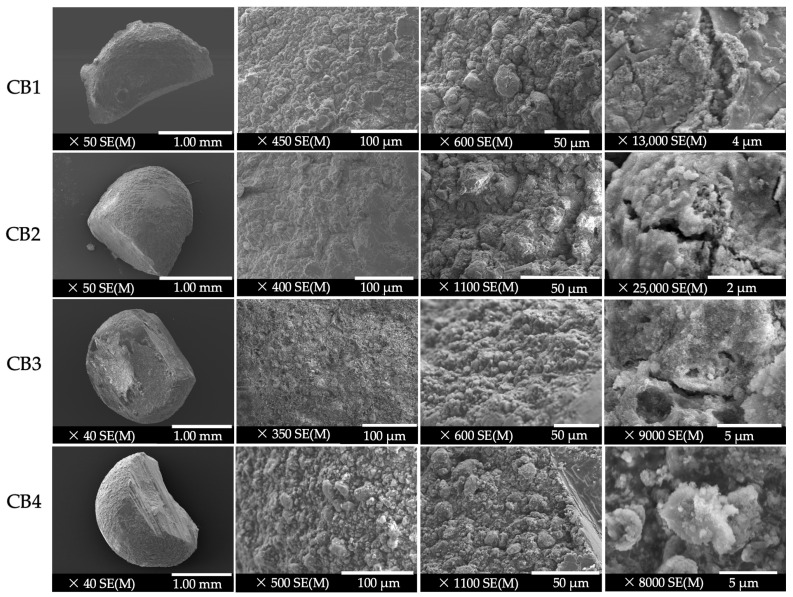
FESEM micrographs of the four beads and of their cross-section.

**Figure 3 polymers-15-01509-f003:**
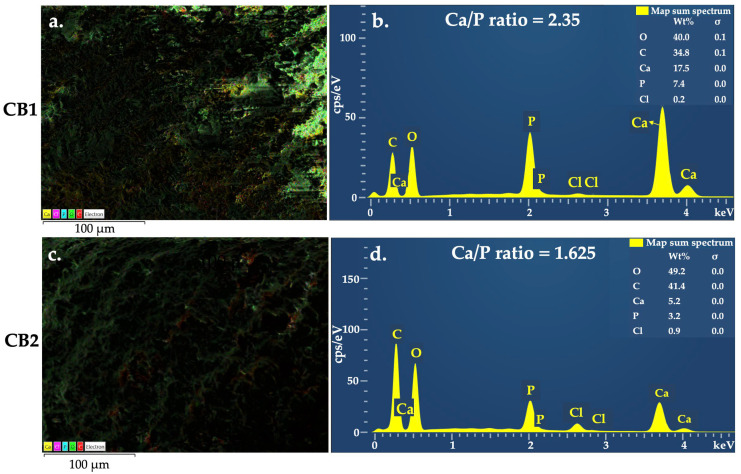
EDX mapping for (**a**) CB1 and (**c**) CB2 and Elemental analysis for (**b**) CB1 and (**d**) CB2.

**Figure 4 polymers-15-01509-f004:**
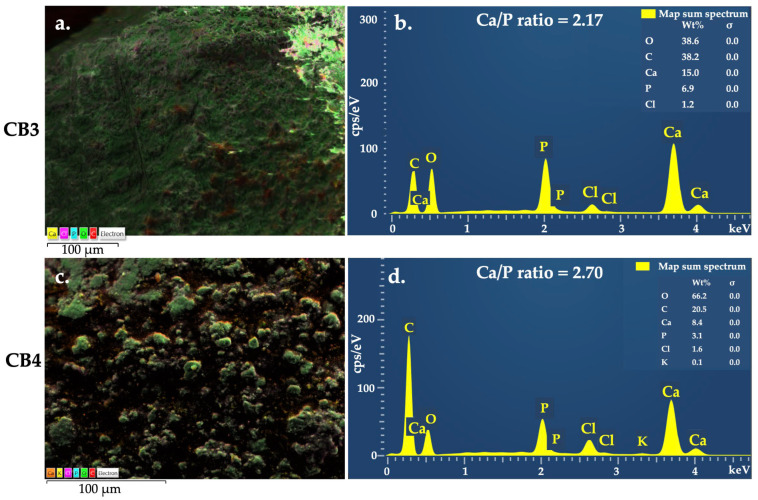
EDX mapping for (**a**) CB3 and (**c**) CB4 and Elemental analysis for (**b**) CB3 and (**d**) CB4.

**Figure 5 polymers-15-01509-f005:**
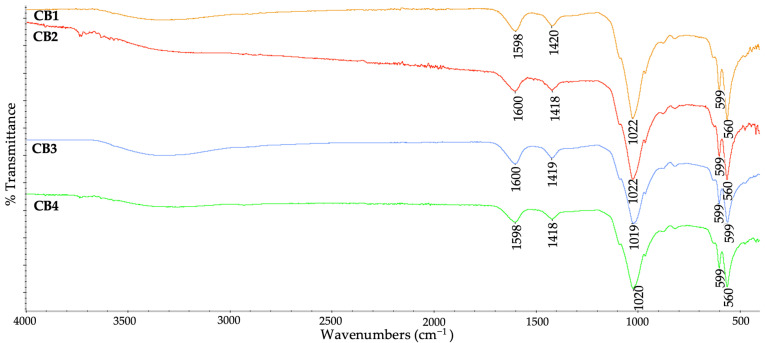
ATR–FTIR spectra of the CBs.

**Figure 6 polymers-15-01509-f006:**
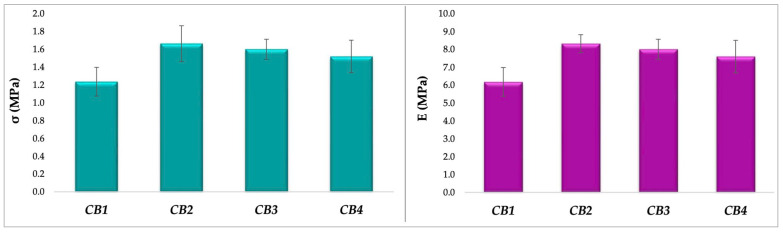
CBs mechanical features: compressive strength—σ (**left**) and Young’s Modulus—E (**right**).

**Figure 7 polymers-15-01509-f007:**
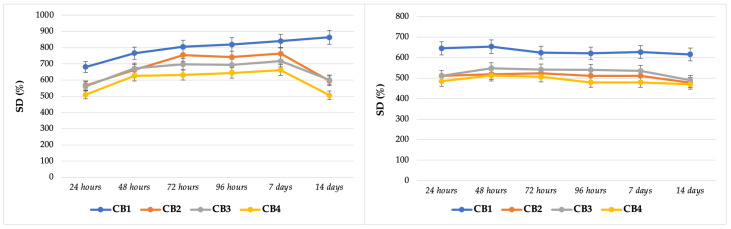
Swelling in HBSS (**left**) and DMEM (**right**).

**Figure 8 polymers-15-01509-f008:**
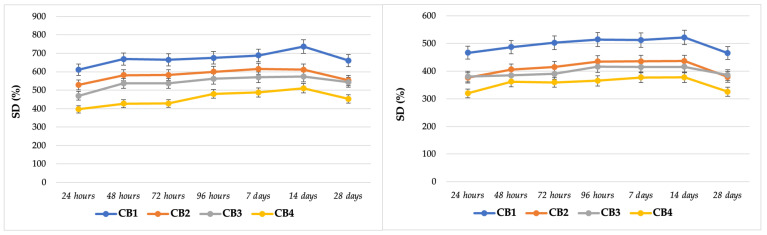
Swelling in physiological serum (**left**) and SBF (**right**).

**Figure 9 polymers-15-01509-f009:**
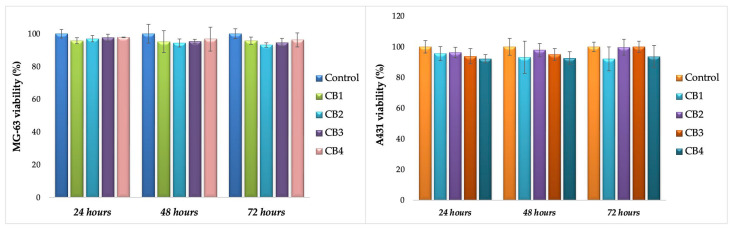
MG-63 viability (**left**) and A431 viability (**right**) after direct contact with CBs.

**Figure 10 polymers-15-01509-f010:**
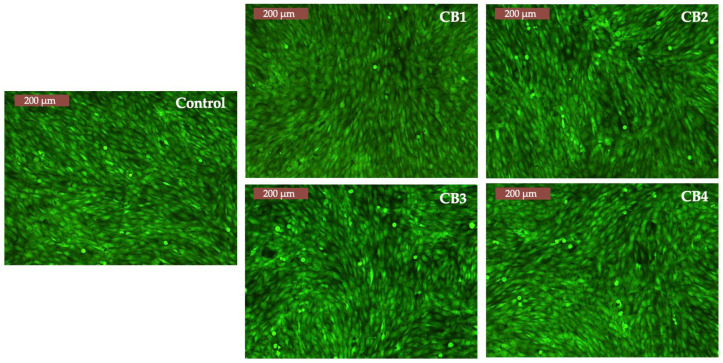
Fluorescent images of MG-63 cells in contact with CBs (72 h) Inverted Phase Contrast Microscope (Leica, Germany).

**Table 1 polymers-15-01509-t001:** CBs codification and composition.

No.	Beads Codification		Beads Composition		
		ALG (wt.)	GG (wt.) ^(a)^	CMGG (wt.) ^(a)^	HA ^(b)^
1	CB1	3%	10%	-	67%
2	CB2		20%	-	
3	CB3		-	10%	
4	CB4		-	20%	

^(a)^ reported to the dry content of ALG; ^(b)^ reported to the dry content of polymers.

**Table 2 polymers-15-01509-t002:** Ca/P ratio calculated from EDX elemental analysis.

CB	Element	Apparent Concentration	wt%	Ca/P Molar Ratio
CB1	Ca	451	17.3	2.35
	P	269	7.3	
CB2	Ca	231	5.2	1.625
	P	202	3.2	
CB3	Ca	860	15	2.17
	P	558	6.9	
CB4	Ca	647	8.4	2.70
	P	351	3.1	

## Data Availability

Not applicable.

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
