# Peer review of "Polysaccharides-Calcium Phosphates Composite Beads as Bone Substitutes for Fractures Repair and Regeneration"

_polymers, 2023, doi:10.3390/polym15061509_

Round 1

Reviewer 1 Report

The authors have mixed different polysaccharides with hydroxyapatite to prepare composite beads. All the chemicals were purchased as ready-made materials and later authors mixed them in different ratios to prepare composites. This lacks scientific novelty in their research. Hence I reject the article.

Author Response

Dear reviewer,

We would like to thank you for your reading and suggestion! Please, accept our opinions, presented in the attached document, regarding your decision to reject our article, due to lack of scientific novelty and to the fact that we have prepared some beads with lack of effort.

Best wishes,
Florina Cojocaru on behalf of the authors

Reviewer 2 Report

The manuscript covers and interesting topic of bio-based substitute for bone structures. However, the manuscript in the current format cannot be accepted for publication. The following major corrections are required before it can be accepted.

1) Introduction:  (only 87 six, based on a quick search on PubMed, using as keywords guar and calcium phosphates 88 [17]): This can not be used a scientific referencing. 

2) The introduction should address how the composites are going to be assessed in line with implantable bone substitutes.

3) Section 3.2: Figure 3 presents the morphology of the synthesized: This should be Figure 2.

4) Figure 2: Please identify scale bars and magnifications on the images.

5) Use the same format to write crosslinking throughout the manuscript. either crosslink or cross-link

6) Figure 3-6: This is abundant. I believe these figures can be added as a supporting document.

7) Moreover, Ca/P  ratio strongly influence mechanical properties of the biomaterials: influences

8)  FTIR: what do we obtain from this FTIR results? are there any sign of ionic crosslinking?

9) Figure 8: Please provide the representative stress-deformation curves.

10) Mechanical Properties: The discussion part is poor. how different formulations led to different mechanical properties? this should be discussed in a stronger manner. how the crosslinking were assessed and evaluated? degree of crosslinking? these should be answered strongly rather than speculations. 

11) Figure 10: The discussion here is also poor. the mechanisms and the reasons for such responses are not addressed properly.

12) conclusion: last line: ccomplex: complex.

13) How about proliferation aspect of the bones?

14) are these composites porous structures? since we know that bones should be porous structure. there was no information on pore density and pore structures.

Author Response

Dear reviewer,

We would like to thank you for your careful reading and constructive suggestions! We have revised the manuscript in accordance with your excellent advises. The point-to-point replies are listed in the attached document. Now we submit the revised manuscript to you for further consideration of publication. Please see more details in our revision. In the revised manuscript we highlighted the modifications using red font. We hope the revised manuscript can satisfy you and meet the high standard requirement for the esteemed Journal.

Best wishes,
Florina Cojocaru on behalf of the authors

Reviewer 3 Report

The authors were presented a capability of organic/inorganic composite bead as bone substitutes. The authors mainly did mechanical and physical characterization of the materials. While, biological analysis using the materials were just biocompatibility test and cell morphology observation. If the authors hope to argue merits of the materials as bone substitutes for repair and regeneration, the data should be added cell proliferation and differentiation, at the least.  In addition, it seems discussion is not sufficient to explain their results in this manuscript.

1. Page 2, line 87. The authors mentioned the novelty of this manuscript. However, if the author wish to present a novelty of this manuscript, the data are required more about the advantages of GG and CMGG as bone substitute materials.

2. What is the difference between water content and interaction with aqueous solution, which the authors mentioned in 2.3.4? Moreover, are DMEM and HBSS possible to be called simulated body fluids?

3. The authors were described amount of beads which the authors utilized for verification of their biocompatibility.

4. The authors should revise table 1 to be clear about bead composition of each groups. 

5. Figure 2-6, the distribution of ceramics in organic part was quite different between GG and MCGG groups. The authors should explain the reason of this phenomenon more. 

6. Page 8, line 264. The authors mentioned CB4 had higher calcium quantity due to stronger interaction between calcium ions and polymer. However, mechanical properties of CB4 was not dramatically higher than other groups. The authors were utilized calcium ions as cross-linker. If CB4 had more stronger interaction with calcium ions, the mechanical properties of the CB4 bead might be higher than other groups. The authors should explain about the correlation between interaction with calcium ions and mechanical properties after ion cross-linking.

7. It seems there is correlation between mechanical properties and water content with aqueous solution. The authors should explain about this.

Author Response

(The authors gave the same response as above.)

Round 2

Reviewer 1 Report

1. Authors need to enhance the quality of Fig. 3. As in current state it is very difficult to see the EDX data values (Fig. 3b).

2. EDX is not a quantitative analysis for determining the Ca/P ration of the samples. As it scans only a specific area of the sample. Authors need to look for other characterization techniques for precise Ca/P ratio analysis.

3. Authors need to enhance the picture quality of Fig. 5 as it get blurred after zooming.

4. Line 314: FTIR spectroscopy shows the obtained chemical structure of the four CBs and the 314 interactions between components. Rewrite the sentence as FT-IR spectroscopy gives information about the presence of functional groups in the sample.

5. Authors must give the wavenumber values for each functional groups noticed in their sample with respect to FT-IR analysis in Fig. 5. Please provide a merged table representing functional groups and their wavenumbers for each composite sample.

6. Line 326 and 327: What do you mean by binding vibration?

7. Line 326: "Also, this high intensity can be attributed to a strong O–H binding vibration". Authors must indicate the value for this high intensity.

8. Include the XRD analysis of all the sample (CaP, AG, GG, CMGG and all the composites) for better comparision before and after composites preparation.

9.State in in abstract which composite was best amongst all and why?

10. Authors need to provide the XRD, FTIR analysis of the samples after interaction with different simulated body fluids. This will provide information as well as compare the samples before and after immersion in the physiological mediums. A quantitative data about the interfacial reactions occured on their surface after interaction with physiological mediums. XRD will show changes in the phases composition whereas FTIR will reveal information about existence of apatite phases. 

11. Authors must provide more experimental details about the shaking conditions of the samples in the subsection "In vitro interaction with simulated body fluids".  Is the physiological medium was refreshed or changed during this study?

12. Why the authors did not studied the interfacial interactions between the organic and inorganic materials? This study is very integral with respect to composites.

13 Authors need to elaborate on the sample preparation for mechanical testing. As there is no information about the shape and dimensions of the samples used in this study. Also provide the mechanical stability data of pure hydroxyapatite. Ceramics are brittle in nature but when mixed with polymers their brittleness reduces. Thus, this information will provide more insights about changing the behavior of a material after preparing composites with polymers.

14. Authors must remove the term CaPs after materials and method section and replace it with hydroxyapatite. This term (CaPs) is written in plural form as a result it creates confusion about which different types of calcium phosphates are being investigated. For eg. check in Fig. A5.

Author Response

Dear reviewer,

We would like to thank you for reconsider your decision and for your careful reading and constructive suggestions! We have revised the manuscript in accordance with your excellent advice. The point-to-point replies are listed in the attached document. Now we submit the revised manuscript for further consideration of publication. Please see more details in our revision. In the second revised manuscript we highlighted the modifications using green font. We hope the revised manuscript can satisfy you and meet the high standard requirement for this esteemed Journal.

Best wishes,
Florina Cojocaru on behalf of the authors

Reviewer 3 Report

.

Author Response

Dear reviewer,

We would like to thank you again for your careful reading and constructive suggestions! Our manuscript was significantly improved after consider them. 

Best wishes,
Florina Cojocaru on behalf of the authors